# Self-Control Buffers the Mortality Salience Effect on Fairness-Related Decision-Making

**DOI:** 10.3390/bs14121121

**Published:** 2024-11-22

**Authors:** Wen Li, Lili Guan

**Affiliations:** 1School of Psychology, Northeast Normal University, Changchun 130024, China; wenli228@nenu.edu.cn; 2Jilin Provincial Key Laboratory of Cognitive Neuroscience and Brain Development, Changchun 130024, China

**Keywords:** mortality salience, self-control, fairness-related decision-making, ego depletion, prosocial behavior

## Abstract

Fairness-related decision-making often involves a conflict between egoistic and prosocial motives. Previous research based on Terror Management Theory (TMT) indicates that mortality salience can promote both selfish and prosocial behaviors, leaving its effect on fairness-related decision-making uncertain. This study integrates TMT with the strength model of self-control to investigate the effects of mortality salience on fairness-related decision-making and to examine the moderating role of dispositional self-control. Participants were primed with either mortality salience or negative affect and then asked to make a series of binary choices (equal allocation vs. unequal allocation favoring themselves) to distribute monetary resources. In both studies, mortality salience heightened selfish tendencies, leading to less equitable monetary allocation. Study 2 further revealed that this effect occurred among participants with low, but not high, self-control. These findings indicate that mortality salience promotes selfishness and inequitable monetary allocation, but that self-control can buffer these effects.

## 1. Introduction

Terror Management Theory posits that the conflict between humans’ survival instinct and the awareness of mortality generates death anxiety, which is mitigated through the pursuit of self-esteem and adherence to a cultural worldview [1,2]. By aligning with a cultural worldview, individuals view themselves as part of something larger, gaining symbolic immortality [2]. Self-esteem reflects the extent to which individuals conform to cultural norms and expectations [3]. Together, cultural worldviews and self-esteem form a dual-component buffer against death anxiety [3]. Previous research has supported these hypotheses across various domains such as religion [4], politics [5], and consumer behavior [6]. However, the effect of mortality salience on fairness-related decision-making remains ambiguous, with contradictory findings in the literature.

In the context of fairness-related decision-making, individuals often experience a conflict between selfish behavior (prioritizing personal gain over fairness) and prosocial behavior (sacrificing personal gain for fairness) [7]. However, research on Terror Management Theory presents mixed findings, suggesting that mortality salience can promote both selfish and prosocial behaviors [8,9,10]. On the one hand, mortality salience can prompt prosocial behavior over selfish behavior. In many cultures, prosocial behaviors are widely endorsed, and prosocial norms become an integral part of people’s cultural worldviews [11]. By adhering to these norms, people can derive a sense of value and security, thereby transcending the fear of death [11]. As a result, mortality salience has been shown to increase prosocial attitudes [9], prompt higher charitable donations [8], and lead to more generous allocation of financial resources [11]. Additionally, mortality salience enhances the value individuals place on fairness and justice [12,13]. On the other hand, mortality salience may also lead individuals to prioritize money over prosocial behavior. Accumulating wealth aligns with cultural norms [14], and money itself holds psychological symbolic power, which can enhance self-esteem [15,16]. Research has shown that contact with money—whether real or play money, or even viewing images of cash—can reduce death anxiety [15]. Mortality salience has been found to increase materialistic desires [6], foster greed [10], intensify the desire for money [16], and increase tolerance for inequity to secure monetary gains [17]. In summary, the impact of mortality salience on fairness-related decision-making appears to be complex and contradictory. We aim to explore this issue by applying the strength model of self-control to Terror Management Theory.

Awareness of mortality evokes fear and anxiety. Aside from attaining a sense of symbolic immortality to reduce mortal concern (e.g., via self-esteem and cultural worldviews), individuals can also suppress death-related thoughts by exerting self-control [18,19]. Self-control varies across states, with an individual’s self-control strength sometimes being stronger and other times weaker [18]. According to the strength model of self-control, tasks requiring self-control, such as thought suppression and impulse inhibition, draw on a limited shared resource [20]. Engaging in such tasks depletes self-control resources, leading to ego depletion, a state in which an individual’s capacity for self-control is reduced, thereby making it more difficult to exert self-control in subsequent tasks [21]. As a result, when individuals exert self-control to suppress death-related thoughts, their resources become depleted, leaving fewer resources available for subsequent tasks requiring controlled processing [18]. For instance, participants performed worse on Stroop tasks, analytical reasoning tasks, and anagram-solving tasks following mortality salience [18]. In summary, mortality salience depletes self-control strength.

More importantly, self-control plays a critical role in regulating selfish and prosocial behaviors [22,23,24,25]. Loewenstein (1996) posits that selfish behaviors, similar to cravings caused by hunger, may be driven by visceral impulses or motivational states [26]. Overriding selfishness requires the exertion of self-control [7,27]. For example, Achtziger et al. (2015) found that participants who experienced ego depletion in a cognitive load task allocated less money to others in a dictator game [27]. Furthermore, as the number of rounds increased, even non-depleted participants became more selfish [27]. Neuroimaging and physiological research also suggest that individuals must exert self-control to suppress selfish tendencies [28,29,30]. For example, Sütterlin et al. (2011), using heart rate variability as a measure of inhibitory capacity, found that self-control is crucial in resisting economic temptations and adhering to fairness norms [29]. Additionally, Knoch et al. (2006) found that after using low-frequency repetitive transcranial magnetic stimulation to disrupt the right dorsolateral prefrontal cortex (a brain region associated with self-control) in the participants, they experienced greater difficulty resisting monetary temptations [28]. In summary, egoistic motive may be the “default motive”, and people need to exert self-control to suppress this tendency [31,32]. Therefore, individuals in a state of ego depletion are less able to suppress egoistic impulses and tend to focus more on monetary rewards [27]. Given that mortality salience depletes self-control [18], we hypothesize that it may impair the ability to suppress egoistic motives, leading to more selfish monetary allocations. However, an alternative hypothesis can also be proposed. Some studies suggest that prosocial motives may, in certain situations, serve as the “default motive” [33,34,35]. For instance, Halali et al. (2013) found that reciprocal behaviors operate more automatically than selfish behaviors and tend to increase when cognitive resources are depleted. In contrast, when individuals focus on maximizing self-interest, reciprocal behaviors are actively inhibited [34]. In fact, the debate regarding default motives has been ongoing [32,36]. Therefore, we propose that two competing hypotheses emerge: mortality salience activates individuals’ default motives, which could either be egoistic or prosocial.

In addition to state differences, self-control also varies across individuals, with some showing greater self-control than others, indicating differences in dispositional self-control [18]. Previous research suggests that individuals with high dispositional self-control are better at regulating negative emotions and exhibit higher prosociality [31,37,38]. Moreover, self-control plays a crucial role in buffering against death anxiety [18,39]. Those with high dispositional self-control generate fewer death-related thoughts after mortality salience priming and exhibit fewer worldview defense responses [18,39]. This suggests that individuals with stronger self-control are more adept at managing death-related thoughts and are less likely to experience ego depletion after mortality salience priming. Therefore, we hypothesize that dispositional self-control moderates the effect of mortality salience on fairness-related decision-making. After mortality salience priming, individuals with low dispositional self-control will have more difficulty suppressing the “default motive”, while those with high dispositional self-control will not.

To test these hypotheses, we conducted two experiments. Study 1 employed a modified dictator game (DG) paradigm to examine the effects of mortality salience on fairness-related decision-making. In the standard DG, one participant (the “dictator”) is given a sum of money or resources and is asked to decide how much, if any, to share with another participant (the “receiver”), who cannot reject the offer [40]. In the present study, we used a modified version of the DG, which is described in detail in Section 2.1. Study 2 measured participants’ dispositional self-control to investigate whether self-control moderates this effect.

## 2. Study 1

### 2.1. Method

#### 2.1.1. Participants

We recruited 74 Chinese undergraduate and graduate students as paid volunteers. Two participants were excluded for selecting incorrect responses in the catch trials (see Section 2.1.2 for a detailed description of the catch trials), leaving a final sample of 72 participants (31 men, 41 women; *M*_age_ = 21.42, *SD*_age_ = 2.96). Participants were randomly assigned to either the experimental condition (mortality salience priming, *n* = 36) or the control condition (negative affect priming, *n* = 36). All participants provided written informed consent, and the research was approved by the local ethics committee.

#### 2.1.2. Materials and Procedure

The experiment began with a priming task in which participants sat at a computer in a dimly lit laboratory. To demonstrate that the defensive reactions induced by mortality salience are specific to the death threat itself and to rule out the potential influence of negative emotions on the experimental outcomes, we chose negative affect priming as the control condition. Participants were asked to indicate their agreement with 28 statements, each displayed on the screen for 7 s with a 0.5 s interval between stimuli. The priming materials were adapted from previous studies [41,42,43]. The 28 statements in the mortality salience condition focused on death-related topics (e.g., “I will eventually die, which makes me feel pessimistic”), while those in the negative affect condition focused on negative emotions unrelated to death (e.g., “I feel anxious about the future”) (see Appendix A). Immediately after the priming task, participants rated their feelings of closeness and fear to death using three items (e.g., “How close do you feel to death after reading the sentences?”, “How unpleasant do you feel about death?”, and “How fearful are you of death?”). Responses were made on an 11-point Likert scale ranging from 0 (not at all) to 10 (extremely).

Next, participants completed a delay task, in which they were required to determine whether 40 calculations resulted in odd or even numbers. Each equation was presented for 7 s with a 0.5 s interval between consecutive calculations. The purpose of this delay task was to shift death-related thoughts from consciousness to a nonconscious yet accessible state (i.e., the fringes of consciousness) [2]. Previous research suggests that only when such thoughts are on the fringes of consciousness do individuals exhibit defensive reactions to mortality salience [2,18,39,44].

The final task was the monetary allocation task, which employed a binary-choice version of the dictator game paradigm [45,46]. Participants were told they had been randomly assigned the role of Proposer, responsible for allocating monetary points, while Receivers were purportedly waiting in another room. Monetary points refer to virtual units used in the experiment, which can be exchanged for real money. Specifically, 1 monetary point is equivalent to RMB 1. In reality, all participants played as Proposers, and there were no actual Receivers. Participants made binary choices between two options: an equal option (10 points to both the Proposer and the Receiver) and an advantageous unequal option (e.g., 14 points for the Proposer and 6 points for the Receiver). They completed 25 such decisions, including two catch trials to identify inattentive participants. In these catch trials, the options presented were both equal but differed significantly in monetary payoff. In the first catch trial, participants were presented with the options “both receiving 10 points” vs. “both receiving 18 points”; in the second, the options were “both receiving 10 points” vs. “both receiving 2 points”. These options were unambiguous and designed to identify inattentive participants. Specifically, a rational and engaged participant would naturally choose the option with the higher monetary payoff for both parties—“both receiving 18 points” in the first catch trial and “both receiving 10 points” in the second. In contrast, failing to choose these objectively better options suggests random responses or a lack of task engagement, because no reasonable interpretation of fairness or self-interest justifies selecting the lower payoff option in these trials. Participants who failed to choose the higher monetary payoff option in both catch trials were excluded from the analysis. To further ensure engagement, participants were informed that one of their decisions (from any of the 25 trials) would be randomly selected to determine partial payment for both themselves and their co-player. At the conclusion of the experiment, participants were fully debriefed, and the deception used in the study was explained in accordance with ethical guidelines.

### 2.2. Results

#### 2.2.1. Subjective Reports During the Priming Task

As a manipulation check, independent-sample t-tests were conducted to examine the effects of priming type (mortality salience vs. negative affect) on participants’ feeling of being close to death, unpleasant emotion, and fearful emotion (see Table 1). We only found a significant difference in feeling close to death (*t*(70) = 4.187, *p* < 0.001, *d* = 0.987), with participants in the mortality salience group reporting feeling closer to death than those in the negative affect group. No significant differences were observed between the two groups in terms of fearful emotion (*t*(70) = 0.477, *p* = 0.635, *d* = 0.112) or unpleasant emotion (*t*(70) = −0.691, *p* = 0.492, *d* = −0.163). Therefore, the manipulation was deemed effective.

#### 2.2.2. Advantageous Inequity on the Monetary Allocation Task

To investigate the effect of mortality salience on fairness-related decision-making, we first calculated advantageous inequity based on participants’ choices in the monetary allocation task. Advantageous inequity was operationalized as the average difference between the Proposer’s and the Receiver’s monetary points [47], where higher values indicated a stronger preference for inequitable distributions favoring the self. Independent-sample *t*-tests were then conducted to examine the effect of priming type (mortality salience vs. negative affect) on advantageous inequity. A significant difference emerged (*t*(70) = 2.329, *p* = 0.023, *d* = 0.549). Participants in the mortality salience condition exhibited significantly higher advantageous inequity (*M* = 5.31, *SD* = 3.45) than those in the negative affect condition (*M* = 3.45, *SD* = 3.31). These results suggest that mortality salience priming, compared to negative affect priming, led participants to allocate more money to themselves at the expense of fairness (see Figure 1).

### 2.3. Discussion

The results of Study 1 demonstrate that mortality salience increased selfish behavior, with participants exhibiting more inequitable monetary allocation in the dictator game. This suggests that egoistic motives may be the default motives for participants. Suppressing death-related thoughts impairs their self-control, making it difficult to inhibit these motives. To further investigate the role of self-control in this process, Study 2 incorporated a measure of participants’ trait self-control. We hypothesized that mortality salience would increase advantageous inequity only among participants with low self-control, but not among those with high self-control.

## 3. Study 2

### 3.1. Method

#### 3.1.1. Participants

We recruited 98 Chinese undergraduate and graduate students as paid volunteers; 4 participants were excluded for failing the catch trials, leaving a final sample of 94 participants (41 men, 53 women; *M*_age_ = 21.21 years, *SD*_age_ = 2.22). In total, 48 participants were assigned to the mortality salience priming group, and 46 to the negative affect priming group. Written informed consent was obtained from all participants, and the study was approved by the local ethics committee.

#### 3.1.2. Materials and Procedure

Prior to the priming task, participants completed the Self-Control Scale [48]. We used the brief version, which consists of 19 items (see Appendix A). Example items include, “I am good at resisting temptation” and “People can count on me to keep on schedule” (1 = not at all like me, 5 = very much like me; *M* = 57.74, *SD* = 10.89, *α* = 0.87). Participants then completed the same priming, delay, and monetary allocation tasks as in Study 1.

### 3.2. Results

#### 3.2.1. Subjective Reports During the Priming Task

As a manipulation check, independent-sample *t*-tests were conducted to examine the effects of priming type (mortality salience vs. negative affect) on participants’ feeling of being close to death, unpleasant emotion, and fearful emotion (see Table 2). We only found a significant difference in feeling close to death (*t*(92) = 6.678, *p* < 0.001, *d* = 1.378), with participants in the mortality salience group reporting feeling closer to death than those in the negative affect group. No significant differences were observed between the two groups in terms of fearful emotion (*t*(92) = −0.943, *p* = 0.348, *d* = −0.194) or unpleasant emotion (*t*(92) = −1.175, *p* = 0.243, *d* = −0.242). Thus, the priming manipulation was effective.

#### 3.2.2. Advantageous Inequity on the Monetary Allocation Task

To investigate whether self-control could buffer the effect of mortality salience on advantageous inequity, we first calculated advantageous inequity from participants’ choices in the monetary allocation task, as described in Study 1. We then conducted a moderation analysis using Hayes’ PROCESS, Model 1 [49]. The independent variable was dummy-coded (0 = negative affect, 1 = mortality salience), and the self-control was mean-centered. Mortality salience had a positive effect on advantageous inequity (*b* = 1.662, *SE* = 0.701, *t* = 2.372, *p* = 0.020), but no significant main effect of self-control was found (*b* = 0.015, *SE* = 0.048, *t* = 0.310, *p* = 0.758). The self-control and mortality salience interaction was also significant (*b* = −0.155, *SE* = 0.065, *t* = −2.390, *p* = 0.019). Simple slope analysis indicated that mortality salience increased advantageous inequity at low levels (−1 *SD*) of self-control (*b* = 3.355, *SE* = 0.989, *t* = 3.391, *p* = 0.001) but had no effect at high levels (+1 SD) of self-control (*b* = −0.031, *SE* = 1.003, *t* = −0.031, *p* = 0.975). Within the mortality salience condition, self-control was negatively associated with advantageous inequity (*b* = −0.140, *SE* = 0.043, *t* = −3.233, *p* = 0.002) but was unrelated to advantageous inequity within the negative affect condition (*b* = 0.015, *SE* = 0.048, *t* = 0.310, *p* = 0.758). These results suggest that mortality salience promotes self-serving monetary allocation behavior, and high self-control can mitigate this effect (see Figure 2).

### 3.3. Discussion

Study 2 found that mortality salience increases selfish behavior, but this effect is restricted to participants with low self-control who exhibited more inequitable monetary allocation in the dictator game. This suggests that participants with low self-control likely depleted their self-control resources while managing death anxiety, leaving them less able to suppress their egoistic motives. In contrast, participants with high self-control retained sufficient resources to regulate these motives effectively.

## 4. General Discussion

The current study examined the impact of mortality salience on fairness-related decision-making and the moderating role of self-control. In Study 1, we found that mortality salience led individuals to become more selfish, as evidenced by their preference for self-serving distribution schemes in the dictator game over fair monetary allocations. Study 2 replicated and extended these findings by measuring trait self-control. We found that the mortality salience effect on fairness-related decision-making occurred among participants with low but not high self-control.

### 4.1. Mortality Salience and Selfish Behavior

Our research suggests that mortality salience increases selfish motivation, leading participants to allocate money more unfairly in the dictator game. One possible explanation is that individuals require self-control to manage death-related thoughts [18]. Since self-control relies on limited cognitive resources, depletion of these resources from one task can lead to ego depletion, impairing the ability to exert further self-control [44]. Consequently, after experiencing mortality salience, individuals may exhaust their self-control resources, making it challenging to suppress their selfish motivations and leading them to prioritize monetary gain over fairness.

However, this explanation assumes that individuals’ default motivation is egoistic. As noted in the introduction, some studies suggest that the default motivation may, in fact, be prosocial [33,34,35]. So why did participants in the present study exhibit more selfish behavior? We speculate that this could be due to the extreme ego depletion caused by mortality salience. Tremoliere et al. (2012) manipulated cognitive load by having participants memorize dot patterns of varying difficulty, categorizing these load levels as low, high, and very high, and compared them with mortality salience priming. They found that mortality salience might be equivalent to a very high cognitive load [50]. Moreover, some research suggests that the relationship between cognitive effort and prosocial behavior follows an inverted U-shape [36,51], implying that extreme ego depletion may reduce prosocial behavior [52]. Therefore, mortality salience priming, akin to a very high cognitive load, may lead individuals to experience extreme ego depletion, reducing their inclination toward prosocial behavior.

Another explanation for why mortality salience makes individuals greedier in the dictator game could be that people try to alleviate death anxiety by acquiring money. Money, with symbolic psychological power, can serve as a buffer against death anxiety [15,16]. Additionally, the drug theory of money suggests that money not only serves as a tool for exchange but can also “act like” natural incentives, similar to drugs, providing emotional regulation and alleviating physical and psychological pain [53]. Since individuals in a state of ego depletion rely more on affective processes rather than cognitive ones [23], those depleted after mortality salience priming may seek to relieve death anxiety through the immediate feedback provided by the “money drug”. Consequently, following mortality salience, individuals tend to allocate more money to themselves at the expense of fairness.

### 4.2. Self-Control Buffers the Mortality Salience Effect

Our findings also highlight that self-control can buffer the mortality salience effect on fairness-related decision-making. Participants with high self-control did not become more unfair in monetary allocation following mortality salience, unlike those with low self-control. Previous research supports that self-control can function as a buffer against death anxiety [18,39,44]. For instance, increased accessibility to death thoughts and worldview defense after mortality salience priming were observed only among individuals with low self-control, and these moderating effects occurred independently of self-esteem [44]. Moreover, temporary enhancement of self-control through glucose consumption has been shown to improve the suppression of death-related thoughts, reducing defensiveness after mortality salience priming [39]. Thus, our results align with these findings, indicating that individuals with high self-control appear to manage death-related thoughts more effectively and retain sufficient self-control resources to suppress their self-serving motivations, thus mitigating the impact of mortality salience on fairness-related decision-making.

An alternative explanation for why self-control buffers the effect of mortality salience on fairness-related decision-making is that individuals with high self-control are more likely to adhere to socially approved norms and have higher levels of self-esteem [44]. Research shows that individuals with high self-control tend to be more disciplined, perform better academically, and are less likely to experience issues with impulse control [38]. Additionally, they exhibit fewer symptoms of psychopathology, demonstrate better psychological adjustment, and tend to have greater empathy, healthier interpersonal relationships, and more fulfilling emotional lives [37,48]. These characteristics enable individuals with high self-control to align more closely with societal expectations, thereby garnering greater respect and social recognition [44]. According to Terror Management Theory, self-esteem and cultural worldviews act as buffers against death anxiety [2]. Individuals with higher self-esteem are better able to cope with mortality salience because they perceive their lives as more meaningful and valued within their cultural context [1]. Given the positive correlation between trait self-control and trait self-esteem [48], individuals with high self-control may be able to buffer death anxiety through self-esteem. Moreover, many cultures, particularly collectivist ones, place a high value on self-control, seeing it as essential for maintaining collective interests and social harmony [54]. Consequently, individuals with high self-control may be particularly protected from the fear of death through a dual-component cultural anxiety buffer, comprising both self-esteem and cultural worldviews, which align closely with the values upheld by their culture.

### 4.3. Limitations and Future Directions

Although our results support our hypothesis that suppressing death-related thoughts depletes limited self-control resources, leading individuals to fail to inhibit selfish motivations after mortality salience priming, more direct and robust evidence is required. Future research could employ neuroimaging techniques, such as fMRI, to examine whether mortality salience affects brain regions associated with self-control, thereby contributing to selfish behavior. Additionally, self-control varies not only as a trait but also as a state [18]. However, we did not examine the role of state self-control. Previous studies have shown that state self-control can influence decision-making and mortality salience effects [18,27]. Future research could explore whether enhancing state self-control can buffer the effects of mortality salience on fairness-related decision-making. Finally, because the participants in this study were all Chinese, a population that values collectivism [54], cultural factors may have influenced the results. In interdependent cultures, individual goals are often achieved through cooperation and adherence to social norms [55]. In such cultures, prosocial behavior is rational, as following prosocial norms through self-control maximizes personal benefits [55]. Therefore, the finding in this study that individuals with high self-control exhibit greater prosociality after mortality salience priming may be specific to interdependent cultures. Additionally, previous research has shown that cultural differences can influence the mortality salience effect [56]. Thus, the differences observed in this study compared to others may be attributed to cultural differences. Future research could investigate whether cultural factors moderate the impact of mortality salience on fairness-related decision-making.

## 5. Conclusions

The present study integrated Terror Management Theory and the strength model of self-control to investigate the effects of mortality salience on fairness-related decision-making and the moderating role of self-control. We found that mortality salience led individuals to behave more selfishly, allocating money less fairly, but self-control buffered this effect. Individuals with low self-control appeared to deplete their limited self-control resources while suppressing death-related thoughts, leading to a failure to inhibit selfish instincts after mortality salience priming. In contrast, individuals with high self-control managed death anxiety and suppressed death-related thoughts more effectively, retaining sufficient self-control resources to regulate selfish motivations. Overall, this study extends both Terror Management Theory and the strength model of self-control by demonstrating that self-control may serve as a key buffer against the effects of mortality salience on fairness decision-making.

## Figures and Tables

**Figure 1 behavsci-14-01121-f001:**
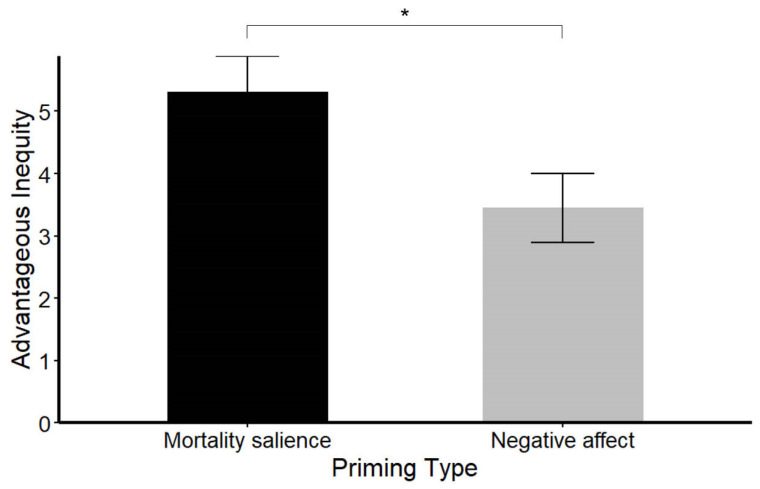
Advantageous inequity under mortality salience condition and negative affect condition. Notes: Error bars represent standard error of mean. * *p* < 0.05.

**Figure 2 behavsci-14-01121-f002:**
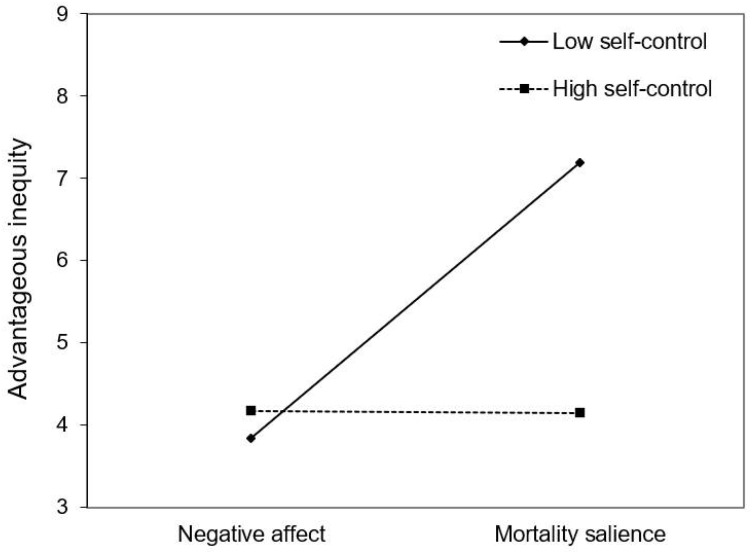
Self-control and mortality salience interaction on advantageous inequity in Study 2.

**Table 1 behavsci-14-01121-t001:** Mean subjective reports during the priming task in Study 1 (*M* ± *SD*).

Priming Type	Feeling Close to Death	Fearful Emotion	Unpleasant Emotion
Mortality salience	4.58 ± 2.70	3.31 ± 2.26	3.01 ± 2.84
Negative affect	2.00 ± 2.54	3.01 ± 2.89	3.53 ± 2.96

**Table 2 behavsci-14-01121-t002:** Mean subjective reports during the priming task in Study 2 (*M* ± *SD*).

Priming Type	Feeling Close to Death	Fearful Emotion	Unpleasant Emotion
Mortality salience	4.66 ± 2.54	2.45 ± 2.15	2.78 ± 2.80
Negative affect	1.44 ± 2.11	2.91 ± 2.62	3.48 ± 2.94

## Data Availability

The data presented in this study are available from the corresponding author upon request.

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
