# Peer review of "Self-Control Buffers the Mortality Salience Effect on Fairness-Related Decision-Making"

_behavsci, 2024, doi:10.3390/bs14121121_

Round 1
Reviewer 1 Report
Comments and Suggestions for Authors
The manuscript investigates the role of self-control as a buffer against the effects of mortality salience on prosocial behaviour. To my mind, the paper offers an interesting contribution to the literature on TMT, as well as self-control and prosocilaity. The experimental designs are well structured and presented, and I believe that with a few minor revisions, the manuscript is very publishable.
Comments on manuscript:
Introduction-
1. The manuscript posits that self-control is a force driving towards prosociality. While the authors do present a good case, several studies have shown that self-control is not inherently prosocial, and may foster selfish behaviour under certain condition (Halali et al., 2014; Uziel & Hefetz, 2014; Schmidt-Barad & Uziel, 2020). The introduction, or at least discussion, should address this complexity.
2. As Study 2 regards trait as well as state self-control, the theoretical distinctions and hypotheses that stem from themn should be included in the Introduction.
Method:
1. Could the authors please elaborate on the nature of "catch trials" and the criteria used for inclusion/exclusion of participants?
2. The experiments utilized negative emotions for the control groups. However, regulating negative emotions may also require self-control resources. Could the authors please elaborate on why negative emotions, rather than a neutral stimulus, were used, and how they differ specifically with mortality salience in relation to self-control and prosociality?
3. A delay task was used to "amplify the effect of mortality salience". Could the authors please elaborate on this? I understand that this is in line with previous research, but I feel it should still be explained within the manuscript.
Results:
1. In Study 2, both a two-way ANOVA and a moderation analysis using Hayes' PROCESS are reported. As the trait self-control is a continuous variable, Hayes' method is more appropriate than ANOVA and should thus be the only analysis reported.
General discussion:
1. In p.7, line 322-324, the authors state that "Consequently, individuals with high self-control 322 may be protected from the fear of death through a dual-component cultural anxiety buffer, 323 composed of self-esteem and cultural worldviews". This statement requires further explanation and ties to the relevant literature.
Reviewer 2 Report
Comments and Suggestions for Authors
This article reports two studies exploring the effect of mortality salience on fairness. The first study shows that mortality salience, compared to negative affect, increases self-interest. The second study reveals that this effect is particularly pronounced for individuals with low dispositional self-control.
The main limitation of this work concerns the theoretical derivation of Hypothesis 1. Overall, I find this article compelling, and I believe it is publishable following suitable revisions.
Detailed Comments:
-
Another useful reference on mortality salience and moral decisions: https://www.sciencedirect.com/science/article/pii/S0010027712001035.
-
The question of whether self-control is necessary to override automatic selfish impulses is more debated than the authors suggest. I urge the authors to discuss this argument in greater detail. For example, as the authors mention, research shows that ego depletion can increase selfishness. Another relevant study on this topic demonstrates that ego depletion may decrease cooperative behavior (https://www.nature.com/articles/srep27219). However, there are also studies indicating that the default choice may actually be cooperative, with cognitive resources needed to calculate a payoff-maximizing strategy (https://www.nature.com/articles/nature11467). A review and meta-analysis on the state of the art in this area, which the authors can find useful for reviewing this topic, can be found here: https://psycnet.apa.org/record/2024-43816-001
-
This discussion requires some significant revisions in the introduction, as the derivation of Hypothesis 1 heavily relies on the assertion that ego depletion increases selfishness.
-
Additionally, I have another concern regarding the introduction. In the initial section, the authors mention that some studies found mortality salience to increase prosociality. However, in the subsequent section, they argue that mortality salience depletes self-control, which is assumed to reduce prosociality. This makes it unclear how the authors reconcile earlier findings that, in some cases, mortality salience appears to enhance prosociality.
-
It would be helpful to include a figure illustrating the results of the first study.
-
“Based on their scores on the Self-Control Scale, participants were further categorized into high and low self-control groups.” How were participants classified? Was a median split used?
Round 2
Reviewer 2 Report
Comments and Suggestions for Authors
The authors have addressed my comments.
Author Response
Thank you for your kind feedback and for acknowledging that we have addressed your comments.